# Site-Specific Proteasome Inhibitors

**DOI:** 10.3390/biom12010054

**Published:** 2021-12-31

**Authors:** Alexei F. Kisselev

**Affiliations:** Department of Drug Discovery and Development, Harrison School of Pharmacy, Auburn University, Auburn, AL 36849, USA; AFK0006@auburn.edu

**Keywords:** ubiquitin-proteasome system, immunoproteasome

## Abstract

Proteasome is a multi-subunit protein degradation machine, which plays a key role in the maintenance of protein homeostasis and, through degradation of regulatory proteins, in the regulation of numerous cell functions. Proteasome inhibitors are essential tools for biomedical research. Three proteasome inhibitors, bortezomib, carfilzomib, and ixazomib are approved by the FDA for the treatment of multiple myeloma; another inhibitor, marizomib, is undergoing clinical trials. The proteolytic core of the proteasome has three pairs of active sites, β5, β2, and β1. All clinical inhibitors and inhibitors that are widely used as research tools (e.g., epoxomicin, MG-132) inhibit multiple active sites and have been extensively reviewed in the past. In the past decade, highly specific inhibitors of individual active sites and the distinct active sites of the lymphoid tissue-specific immunoproteasome have been developed. Here, we provide a comprehensive review of these site-specific inhibitors of mammalian proteasomes and describe their utilization in the studies of the biology of the active sites and their roles as drug targets for the treatment of different diseases.

## 1. Introduction

The ubiquitin-proteasome pathway is the major protein quality control pathway in every eukaryotic cell [1]. The pathway is also responsible for the targeted destruction of numerous regulatory proteins such as cyclin, inhibitors of cyclin-dependent kinases, and transcription factors [2]. Ubiquitylated proteins are degraded by the 26S proteasome, which consists of a 20S proteolytic core and one or two 19S regulatory particles [3,4,5,6]. The 20S core is a hollow cylindrical structure composed of four stacked rings (Figure 1). In eukaryotes, each ring contains seven different subunits. α-Subunits, which are encoded by PSMA genes, form outer rings. These rings form a gated channel that controls substrate access to the proteolytic chamber inside the particle [7]. Inner rings are formed by seven distinct β-subunits, which are encoded by PSMB genes. Two β-rings are identical, and three subunits in each ring carry catalytic residues of the active sites, inhibitors of which are a subject of this review. The β5 subunits are often called “chymotrypsin-like “because they preferentially cleave after hydrophobic residues. The β2 subunits are “trypsin-like” because they cleave after basic residues. Finally, the β1 sites are “caspase-like” because of their ability to cleave after acidic residues. These names reflect similarity in substrate specificity but not in the catalytic mechanism or biological functions. Specific inhibitors of these sites of mammalian proteasomes are reviewed here.

The relative role of these sites in protein degradation was first addressed by site-directed mutagenesis in yeast *Saccharomyces cerevisiae* [8,9,10]. Inactivation of β5(*pre2*) sites caused accumulation of proteasome substrates, increased stress sensitivity, and caused significant retardation of cell growth, despite overexpression of proteasome. Inactivation of β2(*pup1*) sites caused slight growth retardation and reduced degradation rates of specific model substrates. Inhibition of the β1(*pre3*) site did not cause any proteolytic or phenotypic defect, raising an important question of why these sites evolved and were preserved during evolution.

**Figure 1 biomolecules-12-00054-f001:**
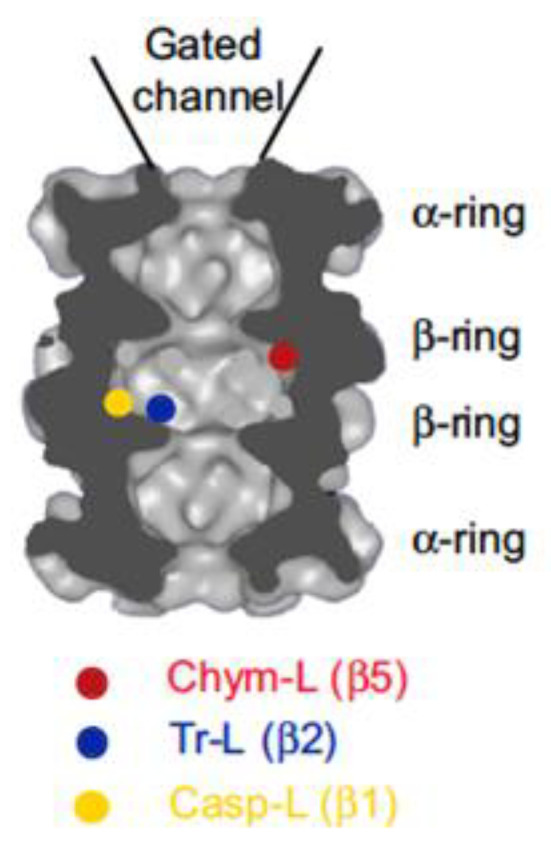
Cross-section of the 20S proteasome showing the location of the active sites. Cut surfaces are dark grey. Active sites are indicated with a colored dot. Chym-L, chymotrypsin-like; Tr-L, trypsin-like; Casp-L, caspase-like. Reprinted from [11], with permission from Elsevier.

First proteasome inhibitors were discovered in the mid-1990s. They were promptly used to demonstrate that proteasomes degrade the majority of proteins in mammalian cells, are essential for the production of MHC class I antigenic peptides [12], and are involved in NF-κB activation [13]. These inhibitors have been indispensable tools to study proteasome biology ever since (reviewed in [11,14,15,16]). Earlier observations that these compounds cause selective apoptosis in the neoplastic cells eventually led to the development and approval of bortezomib for the treatment of multiple myeloma in 2004 [17,18]. Two second-generation inhibitors, carfilzomib and ixazomib [19,20], were approved in the 2010s. Another proteasome inhibitor, a natural product marizomib [21], is undergoing evaluation for the treatment of multiple myeloma and glioblastoma. Multiple myeloma cells are exquisitely sensitive to proteasome inhibitors because the synthesis and secretion of large amounts of immunoglobulins place an enormous load on the protein quality control machinery [22,23,24]. 

All earlier proteasome inhibitors (e.g., MG-132, bortezomib, epoxomicin, *clasto*-lactacystin-ß-lactone, Figure 2) were discovered or developed as inhibitors of the chymotrypsin-like (β5) sites. However, they all co-inhibit other sites at higher concentrations [25,26,27], and the desire to determine whether co-inhibition of β1 and β2 sites is important for the anti-neoplastic activity of these agents was the major driving force behind the development of cell-permeable site-specific inhibitors [28].

Proteasomes are present in all eukaryotes, archaebacteria, and mycobacteria, including *Mycobacterium tuberculosis*. Due to their essential role in the pathogenesis of *M. tuberculosis* and several eukaryotic parasites, e.g., Plasmodia, Leishmania, and Trypanosoma, proteasomes are considered drug targets for the treatment of tuberculosis, malaria, leishmaniasis, African sleeping sickness, and Chagas disease. A comprehensive review of selective inhibitors of mycobacterial and parasitic proteasomes has just been published [29], and this review will therefore focus on the inhibitors of mammalian proteasomes.

In jawed vertebrates, cells and tissues of the immune system predominantly express a different form of proteasome called the immunoproteasome, which features a different set of interferon-γ inducible catalytic subunits (Table 1). The β5c subunit of ubiquitously expressed or “constitutive” proteasomes, encoded by a PSMB5 gene, is replaced in the immunoproteasomes with a β5i (LMP7) subunit encoded by a PSMB8 gene. The PSMB6-encoded β1c is substituted by a PSMB9-encoded β1i (LMP2), and a PSMB10-encoded β2i (MECL1) replaces the PSMB7-encoded β2c subunit. In this nomenclature, “i” indicates an immunoproteasome-specific subunit, and “c” indicates a subunit present only in constitutive proteasomes. Hybrid proteasomes, which contain β5i, β2c, and either β1c or β1i subunits, are also common [30]. Finally, epithelial cells in the cortex of the thymus express thymoproteasomes, which contain β2i, β1i, and a thymus-specific β5t (PSMB11) subunit [31]. We will use β5, β2, and β1 without a suffix only when discussing inhibitors, which block constitutive and immunoproteasome with similar potency.

Immunoproteasomes, while not essential for the survival of mammalian species [32], play an important role in the production of MHC class I antigenic peptides, inflammation, and autoimmunity [33,34,35]. While immunoproteasome involvement in antigen presentation was demonstrated using knockout mice lacking some or all immunoproteasome subunits [32,36,37,38], their critical role in inflammation and autoimmune disease was not fully recognized before the development of ONX-0914 (originally called PR-957), a dual inhibitor of β1i and β5i sites [34]. This inhibitor blocked cytokine production, T-cell activation, and Th17 cells differentiation in numerous murine models of autoimmune disease (see [35] for review). Inhibition of cytokine production was thought to be the consequence of NF-κB inhibition; however, recent work revealed that treatment with ONX-0914 decreases ERK phosphorylation in activated T-cells, most likely due to the stabilization of DUSP6 phosphatase [39]. Similar to multiple myeloma cells, the production of immunoglobulins renders normal plasma cells highly sensitive to proteasome inhibitors [40,41], and proteasome and immunoproteasome inhibitors have shown activity in treating antibody-driven disease (see [35,42] for review). KZ-616, a derivative of ONX-0914 [43], is undergoing clinical trials for the treatment of lupus. 

## 2. Chemical Structures of Proteasome Inhibitors

All active sites of the proteasome and immunoproteasome cleave peptide bonds by the same mechanism, which involves nucleophilic attack of N-terminal catalytic threonine on a scissile bond [7,44]. Most proteasome inhibitors are short N-terminally capped peptides bearing a threonine reactive electrophilic group at the C-terminus (Figure 2) [11,15,16]. Reversible linear and cyclical peptide inhibitors that lack electrophiles have also been developed and are discussed in this review. Peptide-derived inhibitors provide an excellent scaffold for the development of site-specific inhibitors because amino acid side chains can be rationally designed to fit into the specific binding pockets of the active site. Peptide aldehyde MG-132 was the first widely used proteasome inhibitor, but it has been replaced by more specific, potent, and metabolically stable epoxyketones (e.g., epoxomicin, carfilzomib) and boronates (e.g., bortezomib). Epoxyketones react with the amino- and the hydroxyl groups of the N-terminal threonine with the formation of a 6- or 7-membered ring [45,46] and thus take full advantage of proteasome unique catalytic mechanism, ensuring an exquisite specificity. Although it was thought that the chemical nature of the electrophile does not affect site-specificity, replacing epoxyketone with vinyl sulfone increased the specificity of β2 and β5 inhibitors [47,48], albeit at the cost of inhibiting cellular cysteine proteases. The first FDA-approved proteasome inhibitor, bortezomib (Figure 2), is a dipeptide boronate. Initially developed to inhibit serine proteases, peptide boronates inhibit proteasomes with much higher potency [49,50]. The boronate electrophile has also been used to develop site-specific inhibitors. β-Lactones (e.g., *clasto*-lactacystin-β-lactone and marizomib, Figure 2) are the most important group of non-peptide inhibitors. However, attempts to generate site-specific derivatives of these irreversible natural product inhibitors resulted in a modest selectivity [51].

## 3. Site-Specific Inhibitors of the Chymotrypsin-like Sites

Initial efforts to develop β5 inhibitors focused on the selectivity over β1 and β2 and did not distinguish between β5c and β5i (LMP7). Using epoxomicin, an irreversible epoxyketone inhibitor (Figure 2) as a starting point, the Crews group at Yale developed a new epoxyketone YU-101 (Figure 3). It was initially found to have a ~8000 selectivity over other subunits of purified 20S proteasomes [52], but subsequent analysis revealed that it is only ~20-fold selective in cell extracts [28]. Carfilzomib is a derivative of this inhibitor [53]. Carfilzomib and its orally bioavailable analog oprozomib are the most β5-specific of clinically used agents [54] and, are, due to commercial availability, the easiest choices in experiments where specific inhibition of total β5 activity is required. Although they inhibit β2 and β1 sites at higher concentrations or under prolonged exposure [53,55], they can be used as a β5-specific inhibitor if titrated carefully, for example, by treating cells with 30–100 nM carfilzomib for one hour [56,57]. Additional potent β5-selective epoxyketones include NC-005, which is approximately 10-fold more selective than YU-101 [28], and a fluorinated derivative LU-005, which offers a further 10-fold improvement in selectivity [58]. Replacing the epoxyketone warhead in NC-005 with phenyl or methyl vinyl sulfone further increased specificity over β1 and β2 activities [48]. NC-005-VS (vinyl sulfone, Figure 3) does not inhibit β1 and β2 at concentrations as high as 80 μM but is ~10-fold less potent than NC-005 and inhibits cathepsins at concentrations slightly exceeding β5-specific concentrations. Removal of C-terminal electrophiles led to the development of reversible inhibitors. Compound **16** (Figure 3) is a nanomolar inhibitor of β5 that exhibits >100,000-fold selectivity over other sites [59]. In this N- and C-terminally capped dipeptide, bulky hydrophobic C-caping group binds tightly to the S1 pocket of the substrate-binding site, where an amino acid immediately upstream of the scissile bond in the natural substrate binds. The N-caping group occupies the upstream S4 pocket, and dipeptide side chains occupy the S2 and S3 pockets. 

To generate selective β5c and β5i inhibitors, scientists explored slight differences in their substrate specificity, which were revealed by X-ray diffraction [60]. Although both sites cleave after hydrophobic residues, β5c prefers small residues, e.g., Ala, while β5i preferentially cleaves after larger residues, e.g., Trp, Phe, Tyr [59,60]. Inversely, β5i prefers small residues in the P3 position, while β5c has a larger P3 pocket and prefers larger residues (Table 2).

The β5i sites are the prime targets of ONX-0914 (PR-957), the first widely used immunoproteasome inhibitor [34]. It has Phe in the P1 and Ala in the P3 position (Figure 4), while all dual β5c/β5i inhibitors discussed above contain Leu in the P1 and a large hydrophobic residue in the P3 position. ONX-0914 has a modest selectivity over β5c and β1i (Table 3). Replacement of L-Ala in the P3 position with D-Ala and of N-terminal morpholino group with a 3-methyl-H-indene reduced inhibition of β5c and β1i sites, leading to a highly selective β5i inhibitor PR-924 (IPSI) [61]. However, PR-924 is a less potent inhibitor of β5i in a mouse because of subtle differences in substrate binding pockets between murine and human immunoproteasomes [43,62]. Replacement of the phenyl moiety in the P1 position of PR-924 with a cyclohexyl moiety increased β5i selectivity even further, generating LU-015i [63]. Similar manipulation of ONX-0914 generated LU-005i, a potent inhibitor of all immunoproteasome subunits [63]. It is a more potent inhibitor of β2i and β1i than β5i in mouse proteasomes [64]. KZR-329, an epoxyketone originally reported as compound **8** in [43], is more soluble but slightly less selective than PR-924. It also suffers from less potent inhibition of murine β5i. KZR-329 was discovered during the development of KZR-616, a clinical-stage dual β5i and β1i inhibitor with an activity profile similar to ONX-0914 [38]. 

M3258 (Figure 5) is a highly specific boronate inhibitor of β5i that contains a benzofurane moiety in the P1 position [65,66]. This nanomolar inhibitor is almost 1000-fold selective over β5c (Table 3) and does not inhibit any other site at 10 μM. M3258 has anti-myeloma activity in animal models and has entered clinical trials for the treatment of multiple myeloma. Like other β5i inhibitors, it is about ~10-fold less potent against murine β5i. Unlike KZ-616, M3258 is orally bioavailable. Another highly potent β5i specific boronate, compound **22**, was developed by Genentech, alongside compound **8**, an equipotent inhibitor of β1i and β5i [67]. In both inhibitors, β2 residue is replaced with a triazole core.

Several reversible β5i inhibitors do not react with the catalytic threonine (Figure 6a). The Lin laboratory developed a series of N- and C-capped dipeptides, which exhibit the same binding mode as β5 inhibitors described by Millenneum [59,78]. The best characterized of these compounds, DPLG3, shows exquisite 7000-fold β5i specificity in human cells (Table 3), which increases further upon prolonged incubation [68]. In contrast to other β5i inhibitors, it maintains potency and selectivity against murine proteasomes. The incorporation of a β-amino acid into dipeptide inhibitors further increased β5i selectivity, as illustrated by PKS2252 [69]. An attempt to move away from the peptide scaffold led to Asn-ethylenediamine inhibitors [70], in which one of the amino acids was replaced by ethylenediamine, as exemplified by PKS21221 (Figure 6a). This modification retained the potency and maintained selectivity over β1 and β2 sites but reduced selectivity over β5c (Table 3).

β5i subunit contains a unique Cys-54 residue involved in the formation of the S4 and S2 substrate-binding pockets. The Groll group utilized a thiol-reactive chloroacetomide moiety in the P4 position to generate inhibitors that irreversibly react with this cysteine (e.g., CA-4, Figure 6b) [71]. Principia developed PRN1126 and several other reversible inhibitors by introducing a cysteine-interacting moiety in the P2 position [72,79]. In addition, Cys-54 residue appears to play a crucial role in the specific inhibition of β5i by non-peptide oxathiazolones, e.g., HT-2106 (Figure 6c). Originally discovered as inhibitors of the proteasome from Mycobacterium tuberculosis [80], these irreversible inhibitors react covalently with the active site threonine in a manner that requires a Cys-54 facilitated conformational change in the active site to direct the reaction towards the formation of an irreversible adduct [73]. 

Additional non-peptide inhibitors were discovered with the assistance of virtual screening. One of these compounds, a substituted psoralen derivative that fits well in the P1 pocket of β5i, was optimized by altering ring substitutions, including adding different electrophiles designed to turn it into an irreversible inhibitor [74]. N-hydroxysuccinimide and oxathiazolone (e.g., compounds **30** and **42**, Figure 6c) were most potent, with oxathiazolone being more selective than N-hydroxysuccinimide. It is also more potent than HT-2106 and other oxathiazolones derived from mycobacterial proteasome inhibitors [73]. Unfortunately, micromolar concentrations of these nanomolar inhibitors were needed to achieve inhibition inside cells [74]. A series of non-peptide reversible β5i inhibitors were developed by Roche [75]. A structure of the most potent of these compounds, Ro19 (Figure 6d), revealed that the bulky quinoline moiety of this inhibitor occupies the P1 pocket [81], which is larger in β5i than in any other subunits. Interestingly, Ro19 orientation in the active site is perpendicular to ONX-0914. The sole peptide bond of the molecule interacts with the threonine but its orientation is reversed compared to natural substrates, preventing cleavage. Ro19 is a very weak inhibitor of the murine immunoproteasome. Three reviews of immunoproteasome inhibitors have been published recently [82,83,84].

The first β5c specific inhibitors (Figure 7 and Table 4), CPSI (PR-893) and PR-825, by-products of carfilzomib development, were discovered before structural differences between constitutive and immunoproteasomes were elucidated by X-ray diffraction. They are highly selective over β1c and β2c but exhibit only a moderate 10–20-fold selectivity over β1i and β5i [34,61]. Armed with the information generated by the X-ray structure, the Overkleeft laboratory developed two highly selective β5c inhibitors, LU-005c and LU-015c. Both compounds contain Ala in the P1 and a bulky bis-cyclohexyl-Ala side chain in the P3 [85]. They were originally described as a mixture of cis- and trans-stereoisomers in respect to the inner cyclohexane ring (Figure 7). The resolution of these isomers revealed that only trans-isomers exhibit high potency and exceptional selectivity [85]. While highly hydrophobic trans-LU-005c is more potent, it does not inhibit proteasome in cells, probably because it is sequestered in the plasma membrane. The morpholino-capped trans-LU-015c is more soluble and more suitable for cell culture experiments [85].

A non-peptide inhibitor of the β5c subunit, compound **4h** (Figure 5), has also been described [86]. It inhibited β5c sites by a mixed mechanism that changed both K_m_ and V_max_ and was predicted by molecular modeling to bind in the S5 binding pocket. It does not inhibit calpains and cathepsin B. Inhibition of immunoproteasome by this inhibitor was not evaluated.

## 4. Site-Specific Inhibitors of the Caspase-like Sites

The β1 sites were originally called post-glutamyl peptide hydrolase because of their ability to cleave after glutamic acid residues. However, they cleave better after aspartates and were therefore renamed “caspase-like” [87]. Analysis of β1 specificity in the P4-P2 positions using positional scanning libraries with Asp in the P1 revealed that a Pro residue in the P3 position is critically important for selectivity [88]. Peptide aldehyde inhibitors designed based on these screens, Ac-APnLD-al (aldehyde), and Z-PnLD-al, were highly specific but not cell-permeable [88]. Another inhibitor with Pro in the P3, YU-102 (Ac-GPFL-ek, Figure 8), was cell-permeable and β1 specific [89]. Later systematic studies revealed that Pro in P3 was more important for β1 specificity than a P1 residue. Inhibitors with Pro in the P3 were still β1 specific even with Leu in the P1 position [90]. This feature was vital for the development of cell-permeable β1 inhibitors. Replacing an aldehyde derivative of the aspartate in Ac-APnLD-al with an epoxyketone derivative of leucine created NC-001 (Figure 8), which was cell-permeable and more β1 specific than YU-102 [28]. Shortening NC-001 to a tripeptide and the addition of a morpholino cap led to a more potent NC-021 (Figure 8). It inactivates β1 sites in cells at 2.5 μM and maintains specificity at 80 μM, even upon prolonged incubation [56]. It also inhibits β1 in mice (A. Kisselev, unpublished). NC-001 was preceded by a vinyl sulfone HO-282 (Figure 8), which was >10-fold less potent than NC-001 [91]. Another study confirmed that vinyl sulfones are less potent β1 inhibitors than epoxyketones [92]. This work also led to compound **5**, another β1 specific epoxyketone, which has similar potency [92]. It contains a urea fragment in the P3 (Figure 8), initially found in syringolin proteasome inhibitors [93]. 

YU-102 and az-NC-001, an N-terminal azido-derivative of NC-001 developed to simplify conversion of NC-001 into a variety of activity-based probes [28,76], served as starting points for the development of β1i and β1c-specific inhibitors (Figure 8). Replacing Leu with Asp in az-NC-001 generated a potent, albeit cell-impermeable β1c inhibitor LU-001c [77]. A substituted pyridine-γ-lactone, compound **10** (Figure 8), is a specific non-peptide β1c inhibitor with sub-micromolar potency [94]. Although the ability of this compound to inhibit proteasomes inside cells was not reported, its structure suggests that it is much more likely to be cell permeable than LU-001c. Molecular modeling suggested that γ-lactone moiety forms hydrogen bonds with the catalytic threonine but does not react with it.

Replacing the Leu side chain in the P1 of az-NC-001 with a cyclohexyl-Ala moiety, together with the utilization of a 3,3-difluoro-Pro in the P3 position, led to the development of β1i selective LU-001i [63] (Figure 8, Table 5). Replacing a caping group in YU-102 generated a modestly selective DB-310 [95]. These inhibitors are cell-permeable and active in vivo. DB-60 is a macrocyclic derivative of DB-310 with improved metabolic stability and the ability to cross the blood-brain barrier for the potential treatment of Alzheimer’s disease (see below) [96]. A boronate ML604440 [97] and epoxyketone KZR-504 [98] are cell-permeable inhibitors with similar potency and specificity to LU-001i (Table 5). Neither inhibitor has a Pro in the P3 position. However, the 2-pyridone N-terminal capping group in KZR-504, occupying the P3 site is critically important for the β1 specificity. Many of these inhibitors have a bulky hydrophobic residue in the P1 position, which are essential for β1i over β1c selectivity [77,98], indicating that the β1i site is not “caspase-like” in its specificity. Prior to the development of these highly specific inhibitors in the 2010s, UK-101 [99] was used in some studies as a β1i inhibitor (see below); however, it is only 10-fold selective over β5c and cannot be considered β1i-specific in cells that co-express constitutive and immunoproteasomes. 

## 5. Specific Inhibitors of the Trypsin-like Sites

The first β2-specific vinyl sulfone inhibitors, Ac-YRLN-vs and Ac-PRLN-vs, were developed as a result of the first-ever analysis of proteasome specificity using positional scanning libraries [100]. All inhibitors in these libraries contained Asn in the P1 position to facilitate beads attachment during solid-phase synthesis. It was used to explore the influence of amino acids in the P2-P4 positions on the potency of inhibition of all three sites. Although β2 sites cleave after basic residues, inhibitors with Asn in the P1 position were β2 specific if they contained a basic residue in the P3 position [100]. These compounds did not inhibit proteasomes in cells and did not receive wide use. Another selective inhibitor of β2 sites, MalβAla-Val-Arg-al, targeted a unique feature of the S3 pocket in the β2 subunit, a Cys residue in position 118. N-ethylmaleinimide moiety at the N-terminus of this aldehyde reacted irreversibly with this Cys [101].

In developing the first cell-permeable β2 specific inhibitor, we took advantage of leupeptin (Ac-Leu-Leu-Arg-al), a cell-permeable peptide aldehyde inhibitor of intracellular thiol proteases, which is β2 specific in the context of purified proteasomes [25]. Replacing the aldehyde with the proteasome-specific epoxyketone electrophile generated NC-002 (Ac-Leu-Leu-Arg-ek (epoxyketone)), the first cell-permeable β2 specific inhibitor [57]. 3-hydroxy-2-methyl-benzoyl(HMB)-Val-Ser-Leu-vinyl ester was reported as a β2-specific inhibitor [102] but turned out to be inactive when re-synthesized by different laboratories [48]. However, attaching the left-handed peptide fragment of this inhibitor to an epoxyketone derivative of Arg produced a cell-permeable β2 inhibitor NC-022 (HMB-Val-Ser-Arg-ek), which was more potent than NC-002 [57]. Epoxyketone derivatives of Arg are synthetically challenging, have sub-optimal cell permeability, and are not very stable, probably due to intramolecular reaction of guanido group with an epoxy ring (A. Kisselev, unpublished). Replacing Arg-ek with 4-aminomethyl-Phe-vinyl sulfone improved potency, stability, cell permeability, and simplified synthesis, generating LU-102 (Figure 8), the most specific and potent cell-permeable inhibitor of β2c and β2i sites [47]. This study also found that, opposite to the observation with β1 inhibitors, vinyl sulfones inhibited β2 with higher potency and selectivity than epoxyketones. LU-102 has a >100-fold selectivity over β5 and β1 sites (Table 6). At 1–3 μM, it inhibits β2 activity in cells by more than 90 percent after a one-hour treatment. Although LU-102 inhibits cathepsins at slightly higher concentrations, the impact of this off-target inhibition on biological effects of LU-102 can be ruled out using a class-specific cell-permeable inhibitor of thiol proteases E-64d [47]. A sulfonyl fluoride derivative of LU-102 has been reported but has not been tested for cell permeability, inhibition of serine proteases, and suffered from rapid hydrolysis [103]. 

As discussed above, basic residues in the P3 position are important for β2 specificity. NC-012 (Ac-Arg-Leu-Arg-ek) and LU-112, an LU-102 analog with 4-aminomethyl-Phe in the P3 position, are more potent than NC-002 and LU-102 in extracts [47,57]. However, they are weak inhibitors of the intracellular proteasome, most likely because of decreased permeability caused by a second basic group. These compounds served as starting points for the generation of β2-specific activity-based probes (see below). A synthetic analog of a cyclical peptide natural product TMC-95a that contained guanidino-groups in P1 and P3, BIA-IIa (Figure 9), specifically binds to the β2 sites in the X-ray structure of yeast proteasomes [105]. Unfortunately, the authors did not provide kinetic data to quantify β2 specificity, and there is no literature on the inhibition of human and murine proteasomes and cell permeability.

The development of β2c and β2i-specific inhibitors was challenging because substrate-binding pockets of these sites are identical [60]. Nevertheless, LU-002c and LU-012c (Figure 9) were discovered as by-products of efforts to improve the potency and pharmacological properties of LU-102 [104]. They are approximately 10-fold selective over β2i, inhibit β2c in extracts with similar potency as LU-102 (Table 6) but are 4-6-fold less potent inhibitors in intact cells.

During the development of β5i-specific LU-015i (Figure 6), the Overkleeft group noticed that cyclohexyl-alanine in the P1 position improves β2i over β2c selectivity. They replaced it with a bulkier 1-decyl-alanine residue to create β2i inhibitor LU-002i (Figure 8), an epoxyketone that, although 10-fold less potent than LU-102, is approximately 70-fold selective over β2c (Table 6) [104]. LU-002i was originally described as a mixture of diastereomers in the P1 side chain [77], but subsequent analysis revealed that compound **87** (Figure 9) is the active diastereomer. It was also a potent inhibitor of β2i inside cells [104]. A slightly more potent but less specific epoxyketone, compound **10**, was generated during KZR-616 development [43]. Like many β5i-specific inhibitors, it is less potent and specific in mice [43]. Inhibition of murine proteasomes by LU-002i and its diastereomers has not been explored. 

Natural products glidobactin A and C (Figure 10) of the syrbactin class of irreversible proteasome inhibitors co-inhibit β5 and β2 sites in human proteasomes with a low nanomolar potency [106]. Glidobactin C is cell-permeable. Interestingly, these compounds do not inhibit β2c in mice. The Groll laboratory reported that an epoxyketone Ac-Leu-Ala-Ala-ek is an equipotent inhibitor of β5c and β2c [90]. A potent cell-permeable dual β5i/β2i inhibitor, compound **39** (Figure 10), was a by-product of LU-002i development [104]. 

A modestly potent (IC_50_~50–100 μM) compound **1** [107] is interesting because it is a dual inhibitor of β2 and β1 with a novel binding site at the interface of β1 and β2 subunits. This site covers all substrate-binding pockets downstream of the scissile bond in the β2 subunit, to which no other inhibitor binds, and stretches all the way to the S4 pocket of the β1 subunit. 

## 6. Converting Subunit-Specific Inhibitors into Activity-Based Probes

Some site-specific epoxyketones and vinyl sulfones described in this review were converted into fluorescent activity-based probes (ABPs), which were then used to measure the activity of subunits by fluorescent imaging of protein gels [76,77]. These reagents allow rapid assessment of inhibition of individual active sites in crude extracts of various biological samples. The Kim laboratory has developed β1i and β5i probes by introducing a Cy3 and Cy5 fluorescent moiety in the P2 position of UK-101 and PR-924, respectively [108,109]. Such modification was possible because proteasome active sites do not have a defined P2 substrate-binding pocket [7]. They have also developed β2-selective probes by attaching one of these fluorophores to the N-terminus of NC-012 (Ac-Arg-Leu-Arg-ek), and YU-102-derived β1 probes that contain these fluorophores in the P4 position [110]. The Overkleeft laboratory has used a similar strategy of fluorescent N-terminal modification to generate a three-colored cocktail of probes that contain BODIPY(FL)-LU-112, which affinity labels β2i and β2c subunits, BODIPY(TMR)-NC-005-vs, which labels β5i and β5c sites, and Cy5-NC-001 that modifies β1i and β1c sites. They used it to simultaneously label and access occupancy of all six active subunits in a single sample [77]. A combination of BODIPY(TMR)-NC-005, which labels β5 and β1, and BODIPY(FL)-NC-001, which labels β1 sites, can also be used if the resolution of β5i and β5c bands is not required [76]. These probes were used to determine many of the IC_50_ values reported in Table 3, Table 4, Table 5 and Table 6.

## 7. Using Site-Specific Inhibitors to Study the Role of Active Sites as Drug Targets in Cancer

As discussed in the Introduction, the desire to define the roles of individual active sites as drug targets in cancer was the major driving force behind the early development of the site-specific inhibitors discussed in this review. Several studies that used these inhibitors for this purpose are summarized in Table 7. They all found that specific inhibition of a single active site is insufficient to induce apoptosis of malignant cells in culture, regardless of transformed cell tissue of origin. Notably, specific inhibition of β5 sites, the principal target of all FDA-approved inhibitors, did not trigger apoptosis when cells were pulse-treated with inhibitors for one hour to mimic clinical exposure to the drug [111]. Specific inhibitors of β1 and β2 sites, while unable to induce cell death or even limit cell proliferation, dramatically sensitized cells to sub-toxic concentrations of β5 inhibitors [28,47,48,56,57]. β2 inhibitors caused larger sensitization than β1 inhibitors [56]. Most importantly, the addition of β2 inhibitor LU-102 restored bortezomib and carfilzomib sensitivity in primary cells from multiple myeloma patients that developed resistance to these agents [112]. A combination of β1 and β5 inhibitors cannot overcome resistance [54]. Most of these studies used inhibitors that do not distinguish between constitutive and immunoproteasomes. However, using ONX-0914 and LU-102 in hematologic malignancies where immunoproteasome is predominant led to the same conclusion that two sites must be inhibited to induce apoptosis [63,113,114]. In agreement with these conclusions, the clinical efficacy of carfilzomib appears to correlate with its ability to co-inhibit β1i and β2i sites [55]. Unfortunately, dual β5/β2 inhibitors such as glidobactin C, or a dual β5i/β2i inhibitor, compound **39** (Figure 10), have not yet been tested in these experiments. 

Contrary to the observations discussed in the previous paragraph, several studies have demonstrated a single-agent anti-neoplastic activity of site-specific immunoproteasome inhibitors. One study found that β1i inhibitor UK-101 induces apoptosis in prostate cancer cell lines that express β1i subunits and reduces the growth of xenograft tumors in vivo [115]. However, it induced apoptosis at concentrations that inhibited the β5 activity, and when β1i was knocked down. Another study found the activity of β5i-specific inhibitor PR-924 in multiple myeloma [116]; however, the effect was observed at rather high concentrations (i.e., 3 μM) where it may have inhibited β5c and β1i sites (Table 3); inhibition of individual subunits at cytotoxic concentrations was not reported in the study. 

An even more striking example of the single-agent activity of a highly site-specific inhibitor is an anti-myeloma activity of a recently developed highly β5i-specific boronate M3258 [65,66]. It should be noted that this activity was observed upon continuous treatment, which is pharmacologically relevant for this agent because it can be dosed daily and is capable of continuous suppression of β5i activity in mice [66]. This data is consistent with earlier observations that β5-specific inhibitors can induce apoptosis upon continuous treatment [48,61], which prevents recovery of proteasome activity that occurs after pulse treatments with sub-lethal doses of site-specific inhibitors [56,111]. However, the fact that M3258 was cytotoxic to myeloma cells that co-express β5c and β5i sites [65,66] are still surprising because earlier studies have demonstrated that selective inhibition of either β5i or β5c in such cells is not sufficient to induce apoptosis, even upon continuous exposure to the drug [61]. This discrepancy raises the possibility that an off-target effect can contribute to the anti-myeloma activity of M3258.

## 8. Effect of Site-Specific Inhibitors on Protein Breakdown

Although the exploration of the role of the active sites as drug targets in cancer was the main driving force behind the development of site-specific inhibitors, these agents’ ability to inhibit protein degradation, which is the underlying cause of their cytotoxicity, has also been studied. The ability of bortezomib and epoxomicin to inhibit the breakdown of long-lived proteins in pulse-chase experiments did not correlate with the inhibition of β5 sites, and strong inhibition was observed only at concentrations that co-inhibited β1 or β2 sites [25]. Using ubiquitin conjugate accumulation as a surrogate measure of inhibition of protein breakdown, Weyburne et al. found that 1-hr pulse treatment with carfilzomib at β5-specific concentrations caused transient accumulation of ubiquitin conjugates, which disappeared at later times because of biosynthesis of new proteasomes [56]. Combining carfilzomib with β2 inhibitor LU-102 increased conjugates accumulation rate, blocked biosynthesis of new proteasomes, and subsequent clearance of conjugates. NC-021 had a smaller effect than LU-102. The Driessen and Overkleeft laboratories found that β5-specific inhibitor NC-005 did not cause accumulation of the Ubiquitin(G76V)-GFP fusion protein, a model substrate of the ubiquitin-proteasome pathway. Accumulation was observed when NC-005 was combined with β1 inhibitor NC-001 and/or β2 inhibitor LU-102, and the effect of LU-102 was much stronger than the effect of NC-001 [54]. Thus, co-inhibition of β5 and β2 sites is needed to achieve robust and lasting inhibition of protein breakdown. This conclusion has not yet been verified by direct measurement of protein breakdown in a classical pulse-chase experiment.

## 9. Use of Site-Specific Immunoproteasome Inhibitors to Probe the Involvement of These Active Sites in Autoimmunity, Inflammation, and Antigen Presentation

As discussed in the Introduction, an immunoproteasome inhibitor ONX-0914, a dual inhibitor of β1i and β5i sites, has remarkable activity in multiple murine models of inflammation and autoimmunity [34,35,117]. The Groettrup laboratory has used β5i-specific and β1i-specific inhibitors to determine whether inhibition of β5i sites is sufficient to cause these effects and whether co-inhibition of β1i is required for ONX-0914 biological activity. They found that β5i specific inhibitor PRN1126 had a much smaller impact on cytokine production, differentiation of naïve T-helper cells to Th17 cells, and surface expression of MHC class I molecules in mouse models of colitis and autoimmune encephalomyelitis than ONX-0914 [72]. Combining PRN1126 with a β1i inhibitor LU-001i or ML604440 dramatically increased the effect. While PRN1126 ameliorated disease symptoms mildly, a combination with LU-001i had a much stronger effect. LU-005i, which inhibits all three immunoproteasome subunits, inhibited cytokine secretion and ameliorated autoimmunity in a mouse model of colitis [64]. Scientists at Kezaar reached similar conclusions using β1i-specific KZR-504 and β5i-specific KZR-329 in the collagen antibody-induced arthritis model [43]. They also found that KZR-329 inhibited cytokine production by activated PBMC to a lesser extent than ONX-0914. Either KZR-504 or β2i specific inhibitor, compound **10**, increased inhibition to the level of ONX-0914. A combination of β1i and β2i inhibitors did not have any effect. Combining three site-specific immunoproteasome inhibitors did not increase the anti-cytokine effect but caused cytotoxicity. Thus, inhibition of β5i is necessary but not sufficient to achieve maximal suppression of cytokine production, and co-inhibition of either β1i or β2i is required to achieve the effect [43]. Based on this data, the investigators developed KZR-616 as a dual β1i and β5i inhibitor for the treatment of autoimmune disorders [43]. 

Plasma cells and plasmacytoid dendritic cells are highly sensitive to proteasome inhibitors [40,41,118]. Genentech’s highly specific boronate β5i inhibitor, compound **22**, was not cytotoxic to dendritic cells and plasmablasts, while dual β5i/β1i inhibitors were cytotoxic to these cells [64]. Contrary to these observations, reversible N- and C-capped dipeptide β5i inhibitor DPLG3 blocked TLR9-mediated activation of plasmacytoid dendritic cells and PBMCs, suppressed T-cell proliferation, and promoted cardiac allograft acceptance in mice [68]. These findings are important because proteasome inhibitors are used clinically to prevent antibody-mediated transplant rejection [119]. Other β5i-specific N,C-capped dipeptides specifically induced death of antibody-secreting B-cells and inhibited proliferation of activated T-cells [68,70]. These and other examples of using site-specific immunoproteasome inhibitors to dissect the role of individual active sites in the immune response are discussed in greater detail in an excellent recent review by Basler and Groettrup [42].

LMP7 (β5i), LMP2 (β1i), and MECL-1 (β2i) subunits were originally described as proteasome subunits that are induced by IFN-γ, a cytokine that stimulates antigen presentation [120,121]. Proteasome inhibitors that block multiple sites inhibit cell surface presentation of MHC class I ligands [11]. The Groettrup laboratory used specific inhibitor ML604440 to study the role of LMP2 sites in the presentation of a UTY_246-254_ epitope derived from Y-chromosome encoded HY-antigen, which is not generated in LMP2-deficient mice. They found that the the LMP2 inhibitor ML604440 did not block the generation of the epitope and concluded that the replacement of epitope-destroying β1c with LMP2 (β1i) is more critical for the production of this antigenic peptide than LMP2 activity [97]. This example illustrates a unique advantage of site-specific inhibitors over genetic approaches. Another study demonstrated that β1i inhibitor UK-101 blocked the presentation of an antigen derived from the myelin basic protein, which is associated with autoimmune encephalomyelitis, and alleviated disease symptoms in mice, albeit at doses that caused co-inhibition of β5 activity [122]. These experiments raise the possibility of using site-selective proteasome inhibitors for immunomodulation, whereby the presentation of specific sub-sets of antigens is altered without substantial alterations to the overall immune response [123]. Further studies of the effects of site-specific immunoproteasome inhibitors on the production of individual epitope peptides are needed.

A Tg2576 murine model of Alzheimer’s disease is another example of a single-agent biological activity of β1i (LMP2) inhibitors. Immunoproteasome is involved in the activation of the immune system that contributes to the pathogenesis of Alzheimer’s disease (see [124] for review). The Kim laboratory initially found that dual β1i and β1c inhibitor YU-102 reduced cognitive impairment in Tg2576 mice at doses that caused a mild but specific β1i (LMP2) inhibition [125]. The effect was independent of β-amyloid accumulation but correlated with inhibition of astrocytes and microglia activation. Interestingly, YU-102 caused a stronger inhibition of IL-1α production by LPS-stimulated BV-2 cells than ONX-0914 but was a weaker IL-6 and CCL12 production inhibitor. Two β1i-specific inhibitors with improved blood-brain barrier permeability and metabolic stability, DB-310 and DB-60, exerted the same effect without exhibiting any signs of toxicity [95,96]. These findings raise the possibility of using LMP2 inhibitors for the treatment of Alzheimer’s disease. Based on our past experiences with site-specific proteasome inhibitors, it is highly unlikely that overall rates of protein breakdown will be reduced by selective inhibition of LMP2, suggesting that this compound will not interfere with vital proteasome function in neurons [126,127].

## 10. Conclusions

In summary, a two-decade-long effort by several laboratories has provided investigators with a palette of tools to selectively inhibit individual active sites of the proteasome. These inhibitors have proven invaluable tools for dissecting the biology of these active sites. They contributed to the development of the clinical-stage compound KZR-616 and identified β1i as the target for the treatment of Alzheimer’s disease. These reagents present an opportunity to ask many interesting biological questions. Will there be any tissue-specific effects of site-specific proteasome inhibitors, as seen for LMP2 inhibitors in the model of Alzheimer’s disease? Presentation of how many MHC class 1 ligands can be selectively blocked by a specific inhibitor of one immunoproteasome subunit as has been shown for the myelin basic protein epitope? Conversely, can antigen presentation be stimulated by sub-toxic concentrations of inhibitors of constitutive subunits? Does the contribution of active sites to intracellular degradation depend on the amino acid composition of the substrate as found for the β2-dependent degradation of basic proteins by the purified proteasomes [25]? Given the importance of Pro for β1 specificity, would degradation of Pro-rich proteins be sensitive to β1 inhibitors? A recent study demonstrating that sub-toxic, site-specific doses of NC-005, NC-001, and LU-102 exert subtle but distinct effects on the phosphorylation patterns of key regulatory proteins in myeloma cells [128] suggest the untapped biological potential of these reagents.

## Figures and Tables

**Figure 2 biomolecules-12-00054-f002:**
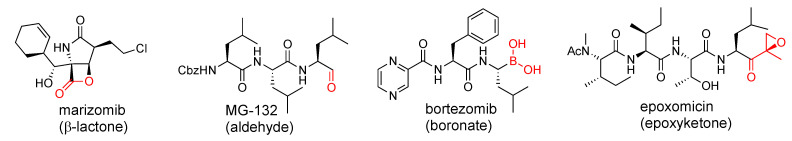
Common proteasome inhibitors. Electrophylic groups reacting with proteasome catalytic threonine are shown in red. Cbz, benzyloxycarbonyl; Ac, acetyl.

**Figure 3 biomolecules-12-00054-f003:**
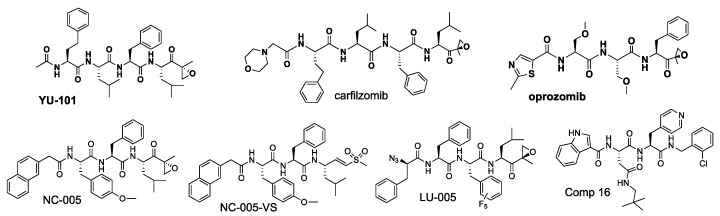
Inhibitors of β5 sites. Commercially available compounds are bold.

**Figure 4 biomolecules-12-00054-f004:**
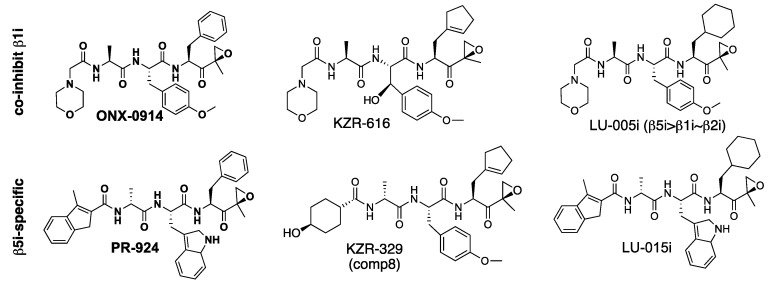
Epoxyketone β5i inhibitors. Commercially available compounds are bold.

**Figure 5 biomolecules-12-00054-f005:**
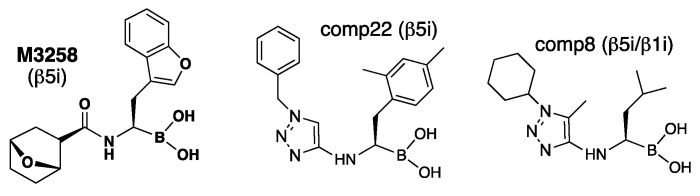
Boronate β5i inhibitors. The commercially available compound is in bold.

**Figure 6 biomolecules-12-00054-f006:**
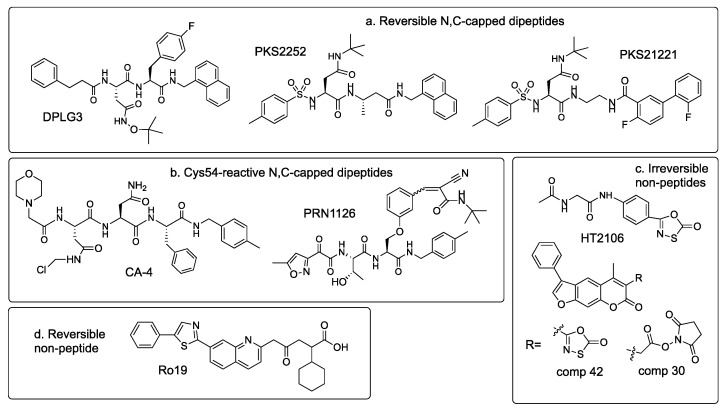
Additional β5i inhibitors. (**a**). Reversible N,C-capped dipeptides. (**b**). Cys54-reactive N,C-capped dipeptides. (**c**). Irreversible non-peptide inhibitors. (**d**). Reversible non-peptide inhibitors.

**Figure 7 biomolecules-12-00054-f007:**
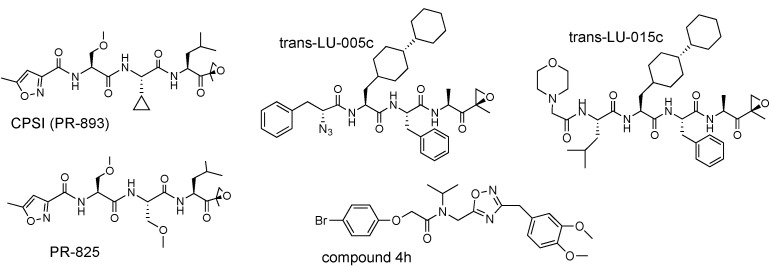
β5c inhibitors.

**Figure 8 biomolecules-12-00054-f008:**
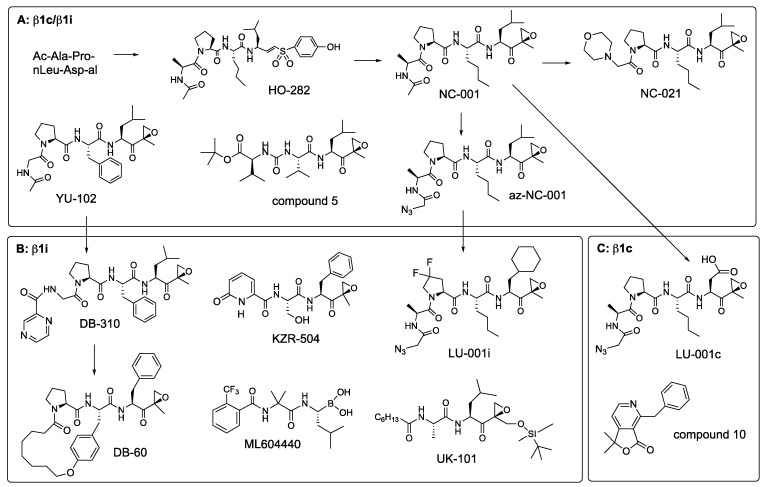
β1 inhibitors. (**A**). Dual β1c/β1i inhibitors. (**B**). β1i (LMP2) inhibitors. (**C**). β1c inhibitors. Arrows show the interrelationship between compounds.

**Figure 9 biomolecules-12-00054-f009:**
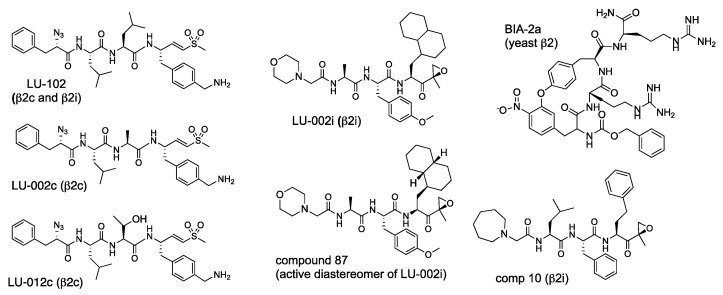
β2 inhibitors.

**Figure 10 biomolecules-12-00054-f010:**
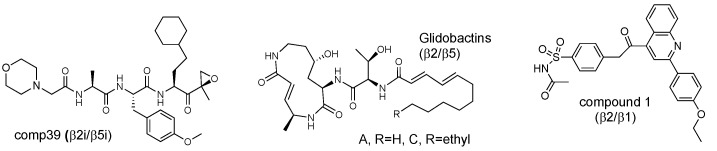
Dual inhibitors of β2/β5 and β2/β1 sites.

**Table 1 biomolecules-12-00054-t001:** The active sites of constitutive and immunoproteasomes.

Activity:	Catalytic Subunit:	
	Constitutive	Immunoproteasome
Chymotrypsin-like (β5)	β5c (PSMB5)	β5i (LMP7, PSMB8)
Trypsin-like (β2)	β2c (PSMB7)	β2i (MECL1, PSMB10)
Caspase-like (β1)	β1c (PSMB6)	β1i (LMP2, PSMB9)

Sites are listed in the order of their relative importance for protein degradation. See Figure 1 for the location of active sites on the proteasome.

**Table 2 biomolecules-12-00054-t002:** Key specificity features of proteasome subunits used to design inhibitors.

Site	Position Relative to Scissile (P1-P1′) Bond	
	P3	P1
β5c	Large hydrophobic	Small hydrophobic (Ala)
β5i	Small hydrophobic	Large hydrophobic (Trp, Phe)
β1c	Pro	Asp, Leu
β1i	Pro	Leu
β2c, β2i	Basic, branched hydrophobic	Basic

Amino acids in the P4 and P2 positions and the positions downstream of the scissile bond are less important for the specificity.

**Table 3 biomolecules-12-00054-t003:** β5i inhibitors.

Compound	IC_50_ *	Fold-Selectivity (IC_50_(βx)/IC_50_(β5i))					Sp	Source	Assay	Ref
	nM	β1i	β5c	β2i	β2c	β1c				
ONX-0914	5.7	9	81	103	192	>1750	h	Raji	ABP	[63]
	39	7.4	11	23	24	>325	h	MOLT4	PrC	[43]
	98	3.3	4	7	9	>250	m	A20	PrC	[43]
KZR-616	39	3.4	18	16	15.5	>270	h	MOLT4	ABP	[43]
	57	3.1	8.4	9	15	>430	m	A20	PrC	[43]
PR-924	25	92	740	>4000	>4000	>4000	h	Raji	ABP	[63]
	39	57	15	>640	>640	>640	h	MOLT4	PrC	[43]
	573	8.3	2.4				m	A20	PrC	[43]
KZR-329	34	54	79	>735	>735	>735	h	MOLT4	PrC	[43]
	374	10.5	7.1				m	A20	PrC	[43]
LU-005i	6.6	45	44	62	378	>1500	h	Raji	ABP	[63]
	160	0.6	19	3	19		h	20S	FS	[64]
	380	0.05	10	0.5	n.i.		m	20S	FS	[64]
LU-015i	8.3	554	855	>1200	>1200	>1200	h	Raji	ABP	[63]
M3258	3.6	>10^4^	694	>10^4^	>10^4^	>10^4^	h	20S	FS	[65]
Comp **22**	4.1	2100	2200			>4800	h	20S	FS	[67]
Comp **8**	1.4	0.81	543			3000	h	20S	FS	[67]
DPLG3	4.5	>7300	7200	>7300	>7300	>7300	h	20S	FS	[68]
	9		1500				m	20S	FS	[68]
PKS2252	5.5		13600				h	20S	FS	[69]
PKS21221	4	>10^4^	27.5	>10^4^	>10^4^	>10^4^	h	20S	FS	[70]
CA-4	0.36	>150	>150	>150	>150	>150	h	20S	FS	[71]
PRN1126	~8	>1000	~30	>1000	>1000	>1000	h	20S	FS	[72]
	~300	>>300	~7	~100	>>300	>>300	m	20S	FS	[72]
HT2106	240	>>40	>40	>>40			h	20S	FS	[73]
Comp **42**	13	>1000	83	>1000	>1000	>1000	h	20S	FS	[74]
Comp **30**	9	630	24	842	995	681	h	20S	FS	[74]
Ro19	25	800	70	800	800	800	h	Ramos	FS	[75]

* IC_50_ values from different reports should not be directly compared to each other because they depend on experimental conditions (e.g., duration of treatment with inhibitor). Sp., species; h, human; m, murine; n.i., no inhibition. “Source” indicates whether purified 20S proteasome or cell extracts were used. Assay methods: FS, fluorogenic substrates; ABP, fluorescent activity-based probe, in-gel detection [76,77]; PrC, ProCISE assay that consists of biotinylated ABP pull-down, followed by subunit-specific ELISA detection [61].

**Table 4 biomolecules-12-00054-t004:** β5c inhibitors.

	IC_50_ nM	Fold-Selectivity (IC_50_(βx)/ IC_50_(β5c))					Species	Source	Assay	Ref
		β5i	β1c	β2c	β1i	β2i				
CPSI (PR-893)	17	21	164	523	13	182	h	Extract *	PrC	[61]
PR-825	20	15	200	500	15	500	h	MOLT4	PrC	[34]
	25	7	70	160	5	70	m	A20	PrC	[34]
LU-005c	75	222	>1300	>1300	>1300	>1300	h	Raji	ABP	[85]
LU-015c	150	52	>666	>666	>666	>666	h	Raji	ABP	[85]
LU-005c-trans	5	2600	>2 × 10^4^	>2 × 10^4^	>2 × 10^4^	>2 × 10^4^	h	Raji	ABP	[85]
LU-015c-trans	30	814	>3570	>3570	>3570	>3570	h	Raji	ABP	[85]
Compound **4h**	37	n.t.	>270	>270			h	20S	FS	[86]

* it is not indicated in the paper extracts of which cell line was used. Abbreviations are explained in Table 3.

**Table 5 biomolecules-12-00054-t005:** β1i inhibitors.

Compound	IC_50_	Fold-Selectivity (IC_50_(βx)/ IC_50_(β1i))					Proteas. Source	Assay	Ref.
	nM	β1c	β5i	β5c	β2i	β2c			
UK-101	104	144	30	10	163	240	Raji	ABP	[63]
LU-001i	95	250	210	210	>100	>100	Raji	ABP	[63]
DB-310	70	8.4	>140	>140			h-i20S *	FS	[95]
DB-60	184	46	2.2	25			h20S	FS	[96]
KZR-504	50	~900	94	135	>4900	>4900	MOLT4	PrC	[98]
ML604440	10		>100	>100			m20S	FS	[97]

h, human; m, murine; * RPMI-8226 extract was used to measure inhibition of c20S.

**Table 6 biomolecules-12-00054-t006:** β2 inhibitors.

Comp	IC_50_, μM						Sp	Source	Ass	Ref
	β2c	β2i	β5c	β5i	β1c	β1i	h	Raji	ABP	[104]
LU-102	0.013	0.020	1.3	1.2	>100	>100	h	Raji	ABP	[104]
LU-002c	0.005	0.14	1.3	2.8	>100	>100	h	Raji	ABP	[104]
LU-012c	0.007	0.11	0.75	2.1	>100	>100	h	Raji	ABP	[104]
LU-002i	12.1	0.22	>100	>100	>100	>100	h	Raji	ABP	[104]
Comp **87**	19	0.19	>100	28	53	>100	h	Raji	ABP	[104]
Comp **10**	0.41	0.071	0.49	0.35	>25	>25	h	MOLT4	PrC	[43]
	0.57	0.31	0.38	1.92	>25	>25	m	A20	PrC	[43]
Comp **39**	2.5	0.057	5.0	0.046	>100	>100	h	Raji	ABP	[104]

Abbreviations are the same as in Table 3. Comp **87** is presumed to be an active diastereomer of LU-002i.

**Table 7 biomolecules-12-00054-t007:** Studies demonstrating the sensitization of transformed cells to β5-specific inhibitors by β2 and β1 inhibitors.

Cell Lines Derived From	β5 Inhibitor *	β1 and β2 Inhibitors	Ref
Multiple myeloma	NC-005	NC-001	[28]
HeLa	NC-005-vs	NC-001	[48]
Multiple myeloma	LU-005, Cfz	NC-001, NC-022	[57]
Multiple myeloma	carfilzomib	LU-102	[47]
Primary ALL	LU-015i	LU-001i	[77]
Bortezomib and carfilzomib-resistant myeloma	NC-005 carfilzomib	NC-001, LU-102 LU-102	[54] [112]
Triple-negative breast, lung, liver, ovarian cancer	carfilzomib	NC-021 & LU-102	[56]
Myeloma	ONX-0914	LU-102	[113]
ALL	ONX-0914	LU-102	[114]

* All treatments with β5 inhibitors were 1 hr-pulse treatments.

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
