# Peer review of "Site-Specific Proteasome Inhibitors"

_biomolecules, 2021, doi:10.3390/biom12010054_

Round 1

Reviewer 1 Report

This manuscript provides a very careful review about the proteasome inhibitors, their development and their use in clinics. It also very nicely recapitulates how these proteasome inhibitors were used to decipher the role of proteasome subtypes in various cellular functions. This is very well written and the detailed analysis of the various proteasome inhibitors with regard to their active site specificity will be very useful to the proteasome community in general. I have only a few comments:

It is certainly not mandatory, but for non proteasome-experts that might grasp some clinical interest in proteasome inhibitors, a schematic figure displaying the basic structure proteasome, their catalytic subunits and their active sites might be useful.

In table 2: there might be a mistake in the content of the table for the b5c and b5i subunits lines. I think that the “(Trp, Phe)” and “(Ala)” displayed in the P3 column of the table might be rather placed in the P1 column: “(Trp, Phe)” after the large hydropobic P1 residue of the b5i line and “(Ala)” after the small hydrophobic P1 residue in the line b5c. (in agreement to what is stated in the text, lines 151 to 156).

Line 164 claims a 20-fold ONX-0914 selectivity for b5i over b5c and b1i. I do not understand how this is calculated with regard to table 3, where the indicated selectivity is 9 and 7.4 for b1i and 81 and 11 for b5C. Could you clarify?

Line 130: “it’s” should be replaced by “its”

Author Response

We thank the referee for a favorable review, and for suggestions to improve the manuscript. 

A new figure 1 displaying overall structure of the proteasome, and location of the active sites has been added.

We have changed Table 2 as suggested by the referee.

ONX-0914 selectivity - since selectivity varies between studies (Table 3) we decided not to mention a specific number in the text.

Line 130 (now line 144) "It's" has been corrected to "its"

Reviewer 2 Report

In this Review, Alexei has done a particularly great job in putting together a thorough, organized and up-to-date summary of a multitude of proteasome inhibitors. Breaking down these inhibitors into groups based on their chemical features and site specificity, and compiling their pharmacological properties in the tables are especially helpful for the readers.

I have only some minor comments/suggestions:

  1. Including X-ray structures of the proteasome active sites as well as co-structures with inhibitors would help illustrate the site specificity and design rationales.
  2. It would be nice to have a small paragraph on recent advances of using site-/species-specific proteasome inhibitors to target infectious/parasitic diseases such as malaria and TB.

Lines 388-389: “Interestingly, these compounds do 388 not inhibit b2c (in) mice.”

Author Response

We thank the referee for a favorable review, and for  the suggestion to use X-ray structures, and have seriously considered including them. However we have concluded that it will make review more difficult to read for pure biologists with little interest in structural biology, and ultimately decrease the readership. Furthermore, adding structures would further increase size of the review which is already longer and contains more figures than suggested by editorial guidelines. Instead, we have accepted the suggestion  of the editor and referee 1 to add a scheme illustrating the location of individual active sites in the proteasome, which is a new Fig.1.

We have mentioned parasitic and mycobacterial proteasome and are now citing a comprehensive recent review of their inhibitors.

There is no "388" between "do" and "not" in lanes 388-389; the review must have mistaken lane number for a part of the text.

Reviewer 3 Report

It was with a quite pleasure to review the review manuscript of "site specific proteasome inhibitors". The manuscript is well organized with accurate historical account and current affair of site specific proteasome inhibitors and use of these inhibitors as tool to interrogate the biology and as therapeutics. I recommend to publish with the correction of following errors.

  1. Page 2 line 64, immunoproteasome is not only found in mammals, jawed vertebrates also express immunoproteasome
  2. Page 2 line 69, a reference could be cited (PMCID: 2972972)
  3. Page 5, figure 3. chirality of P2 in KZR-616 and KZR-329 was not defined
  4. Page 8, figure 6, structure of trans-LU-015c is incorrect. A P4 Leu is missing.

Author Response

We thank the referee for a favorable review, for suggestions to improve the manuscript, and for noticing our errors.  We have added the reference suggested by the referee, and have corrected all errors.